# Comprehensive Characterization of the Structure and Gel Property of Organo-Montmorillonite: Effect of Layer Charge Density of Montmorillonite and Carbon Chain Length of Alkyl Ammonium

**Jun Qiu [1,*], Dongliang Liu [2], Yueting Wang [2], Guowei Chen [2], Shan Jiang [2], Guoqing Li [2], Yaqi Wang [1], Wenxin Wang [1], Peng Wu [1], Xiaodong Liu [1], Guifang Wang [3] and Xianjun Lyu [1]**

[1] College of Chemical and Biological Engineering, Shandong University of Science and Technology, Qingdao 266590, China; wangyaqi0325@gmail.com (Y.W.); 03250108wang@gmail.com (W.W.); wupeng20190108@163.com (P.W.); skd992431@sdust.edu.cn (X.L.); lux032046@gmail.com (X.L.)

[2] College of Safety and Environmental Engineering, Shandong University of Science and Technology, Qingdao 266590, China; 1925706004ldl@gmail.com (D.L.); wangyueting1026@gmail.com (Y.W.); wei771941279@gmail.com (G.C.); jswork20200212@163.com (S.J.); liguoqing201907@gmail.com (G.L.)

[3] School of Resources Environment and Materials, Guangxi University, Nanning 530004, China; wangguifang2019@gmail.com

\* Correspondence: qiujun@sdust.edu.cn; Tel.: +86-159-6694-9378

**Abstract:** In this work, the effect of layer charge density of Na-montmorillonite (Na-MT) and carbon chain length of alkyl ammonium on the structure and gel property of organo-montmorillonite (organo-MT) was studied by using X-ray diffraction (XRD), Fourier transform infrared (FTIR), thermogravimetric (TG) analysis, contact angle test, molecular dynamics (MD) simulation, and gel apparent viscosity determination experiment. The results of XRD show that Na-MT with lower layer charge density is easier to swell after intercalation of alkyl ammonium, and the basal spacing of organo-MT increases with the increase of carbon chain length. The results of FTIR show that the absorption bands at 2924 cm$^{-1}$ and 2853 cm$^{-1}$ shift towards low frequency region with the increase of carbon chain length, and the absorption bands at 515 cm$^{-1}$ and 463 cm$^{-1}$ move towards high frequency region when the layer charge density increases. The mass loss of organo-MT evidently increases with the increase of layer charge density of Na-MT or carbon chain length of alkyl ammonium. The contact angle test results are well in line with the TG data and reveal that alkyl ammonium with longer carbon chain can significantly improve the hydrophobicity of organo-MT. MD simulation indicates that, when the layer charge density is low, the distribution of alkyl ammonium gradually changes from parallel double layers to partially inclined distribution with the increase of carbon chain length, but when the layer charge density is high, the distribution of alkyl ammonium gradually changes from three layers into four layers. The test results of the apparent viscosity of the gel formed by organo-MT in xylene show that the apparent viscosity of organo-MT gel is negatively correlated with the layer charge density of Na-MT and positively correlated with the carbon chain length of alkyl ammonium.

**Keywords:** montmorillonite; layer charge density; alkyl ammonium; carbon chain length

## 1. Introduction

Montmorillonite (MT) is a kind of typical phyllosilicate clay mineral and is rich in nature. Its structure is an octahedral sheet sandwiched by two tetrahedral layers [1,2]. Because of its large specific surface area (SSA), low cost, high cation exchange capacity (CEC), and good swelling

ability in aqueous solution, MT has attracted wide attention both in academia and in industry [3–6]. The occurrence of non-equivalent isomorphic substitution of $Al^{3+}$ for $Si^{4+}$ in the tetrahedral sheets and $Mg^{2+}$ for $Al^{3+}$ in the octahedral layers makes MT layers negatively charged [7]. To balance the negative layer charge density of MT, some hydrophilic exchangeable cations, such as $Na^+$, $K^+$, and $Ca^{2+}$, are usually attracted into the interlayer domain, and due to the hydration of inorganic cation [8,9], confined basal spacing, and high surface energy, natural MT performs badly in organic phase environment [6,10].

In order to expand the application of MT in industry, organo-MT is synthesized by intercalating surfactant into interlayer space of natural MT [11]. In this way, interlayer distance of MT obviously increase and new sorption sites can be formed [12], besides, inorganic cation can be exchanged by surfactant, which results in that the surface property of MT converts from hydrophilic to hydrophobic. As a result, the organo-MT can be applied in many fields, such as polymer MT nanocomposite [13,14], adsorbent for organic contaminant [15,16], oil-based drilling fluid [17], pharmaceutical, cosmetic additives, thixotropic agent in paint, etc. [18].

So far, in the research process of organo-MT, various kinds of cationic [17], anionic [14], nonionic [19], amphoteric, and even hybrid surfactants have been used [5,20–22], and many scholars found that the structure and property of organo-MT were closely related to many factors, such as carbon chain length and concentration of surfactant. Zhang et al. and Ma et al. found that cationic surfactant with long carbon chain could significantly increase the interlayer distance of organo-MT, and organo-MT had better adsorption capacity to organic contaminant than natural MT [12,15]. Yunfei Xi and his co-writers reported that variation in the $d_{(001)}$ was a step function of octadecyl trimethyl ammonium bromide concentration, the arrangement of surfactant between MT layers changed stepwise from monolayer to pseudo trimolecular layer with the increase of concentration of surfactant [7]. Taleb et al. proposed that the thermal stability of organo-MT was better when Na-MT is modified by surfactant with higher concentration and longer carbon chain [23]. Zhuang et al. concluded that organo-MT prepared by surfactant with bigger size and lower Hydrophile-Lipophile Balance value could be easier to exfoliate in oil [17]. Similar rules were also verified by other investigators [24–26].

Moreover, layer charge density of MT, type of interlayer cation, modification condition, and so on can also influence the structure and property of organo-MT. Wu et al. and Chen et al. studied the influence of type of interlayer cation on organic intercalation of MT, and they found that, the weaker the electrostatic attraction between interlayer cation and MT sheets, the easier the organic intercalation of surfactant into MT layers [11,27]. Peng et al. studied the effect of pH on the adsorption of dodecyl amine (DDA) on MT surfaces in aqueous solution and found that the hydrophobicity of DDA-MT was the best when the pH of aqueous solution is 8.0 [28].

Although large amounts of works on the organic modification of MT have been done, but to our knowledge, there is no systemic comparative study about the effect of layer charge density of Na-MT and carbon chain length of alkyl ammonium on the structure and gel property of organo-MT. For this purpose, four types of alkyl ammonium with different carbon chain length are used as intercalated modifier, two kinds of MT with different layer charge density are used as research materials, and the prepared organo-MT under different conditions are comprehensively characterized by XRD, FTIR, TG analysis, contact angle test, and gel apparent viscosity test combined with molecular dynamics (MD) simulation. The research results of this work can bring new insights into the influence mechanism of layer charge density of MT and carbon chain length of alkyl ammonium on the structure and gel property of organo-MT, which can pave the way to apply organo-MT in industry.

## 2. Materials and Methods

### 2.1. Materials

The two kinds of Ca-MT are obtained from Shandong and Inner Mongolia of China, respectively, and are purified by natural sedimentation method. Their chemical compositions are presented in Table 1. Their semi-cell structure formulas are $(Ca_{0.17}K_{0.02}Mg_{0.01})_{0.38} \cdot \{(Al_{1.63}Fe_{0.01}$

$Mg_{0.36})_{2.00}[(Si_{3.98}Al_{0.02})_{4.00}O_{10}](OH)_2\}$ and $(Na_{0.03}Ca_{0.29}K_{0.04}Mg_{0.02})_{0.69}\cdot\{(Al_{1.27}Fe_{0.04}Ti_{0.03}Mg_{0.66})_{2.00}$ $[(Si_{3.94}Al_{0.06})_{4.00}O_{10}](OH)_2\}$, and their corresponding layer charge density is 0.38 and 0.69, respectively. In order to improve the dispersity of Ca-MT in water, both of them were sodium-modified by $Na_2CO_3$ of 4.0% (wt %) for 1.0 h at room temperature, dehydrated by centrifugation in the centrifugal machine (TD5A-WS, Jiangsu Jintan Instrument Factory, Jintan, China), dried in the oven (302, Shandong Longkou Xianke Instrument Co., Ltd., Longkou, China) at 80.0 °C for 24.0 h, and then ground to less than 200 mesh by vibrating grinder (XZM-100, Wuhan Prospecting Machinery Factory, Wuhan, China). The prepared Na-MT were denoted as Na-MT1 and Na-MT2, respectively, and their SSA, pore volume (PV), average pore diameter (APD), and cation exchange capacity (CEC) are measured and showed in Table 2.

**Table 1.** Chemical composition of two kinds of Ca-MT (wt %). MT = montmorillonite.

| Type | $SiO_2$ | $Al_2O_3$ | MgO | $Na_2O$ | $Fe_2O_3$ | $K_2O$ | CaO | $TiO_2$ |
|---|---|---|---|---|---|---|---|---|
| Ca-MT1 | 68.65 | 16.04 | 4.15 | 1.99 | 2.98 | 2.86 | 3.33 | 0 |
| Ca-MT2 | 66.83 | 18.79 | 6.44 | 0.34 | 2.80 | 0.41 | 3.98 | 0.41 |

**Table 2.** Specific surface area (SSA), pore structure, and cation exchange capacity (CEC) of two kinds of Na-MT.

| Types | SSA ($m^2$/g) | PV ($cm^3$/g) | APD (nm) | CEC (mmol/100g) |
|---|---|---|---|---|
| Na-MT1 | 32.6945 | 0.0575 | 7.0370 | 90.69 |
| Na-MT2 | 44.9558 | 0.0632 | 5.6280 | 108.23 |

Both absolute ethyl alcohol and xylene were purchased from Tianjin Fuyu Fine Chemical Co., Ltd. (Tianjin, China). The four types of alkyl ammonium are dodecyl trimethyl ammonium chloride (DTAC), tetradecyl trimethyl ammonium chloride (TTAC), hexadecyl trimethyl ammonium chloride (HTAC), and octadecyl trimethyl ammonium chloride (OTAC), respectively. The DTAC with 98.0% purity was purchased from Shanghai Shanpu Chemical Co., Ltd. (Shanghai, China). The TTAC with 99.0% purity was purchased from Shanghai Macklin Biochemical Co., Ltd. (Shanghai, China). The HTAC with 98.0% purity was purchased from Tianjin Zhiyuan Chemical Reagent Co., Ltd. (Tianjin, China). The OTAC with 99.0% purity was purchased from Sinopharm Chemical Reagent Co., Ltd. (Shanghai, China). All the alkyl ammonium were directly used without further purification, and their molecular structures are shown in Figure 1.

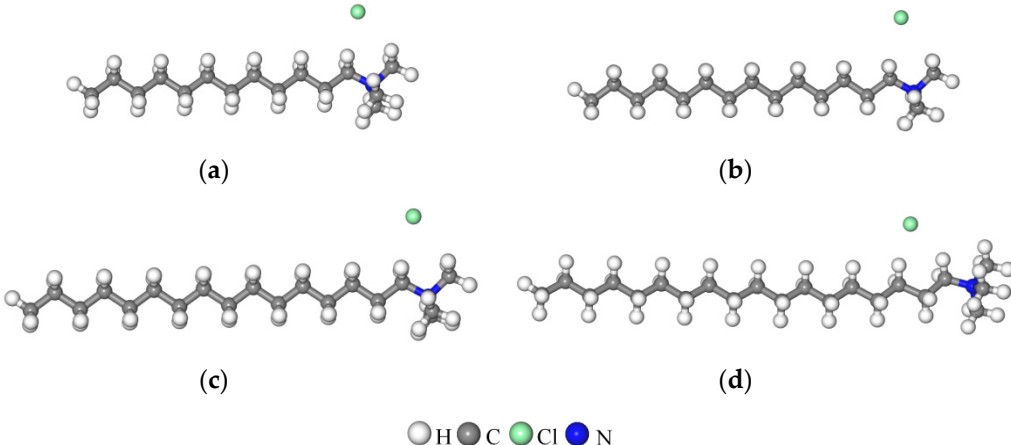

**Figure 1.** The structure of alkyl ammonium, (**a**) dodecyl trimethyl ammonium chloride (DTAC), (**b**) tetradecyl trimethyl ammonium chloride (TTAC), (**c**) hexadecyl trimethyl ammonium chloride (HTAC), (**d**) octadecyl trimethyl ammonium chloride (OTAC).

## 2.2. Preparation of Organo-MT

Organo-MT were prepared by dissolving 10.0 g of Na-MT into distilled water at 80.0 °C in a 250.0 mL beaker, then slowly adding the dispersed solution of DTAC, TTAC, HTAC, or OTAC under the same temperature condition. The mass of alkyl ammonium added was equivalent to 1.0 CEC of Na-MT, and the mass ratio of water to dried Na-MT was 20:1. The mixture was stirred for 2.0 h at 80.0 °C, then the prepared organo-MT was collected by centrifugation at 4000 r/min, washed with distilled water until there is no Cl$^-$ examined by $AgNO_3$, dried in the drying oven (302, Shandong Longkou Xianke Instrument Co., Ltd., Longkou, China) at 80.0 °C for 48.0 h, and ground to less than 200 mesh by vibrating grinder (XZM-100, Wuhan Prospecting Machinery Factory, Wuhan, China). The obtained samples are denoted as $C_n$-MT1 or $C_n$-MT2, where $n = 12, 14, 16$, or 18, respectively.

## 2.3. Analytical Methods

Powder XRD patterns of samples were recorded with a D/max-r B diffractometer using Cu K$\alpha$ radiation at 40.0 KV and 100.0 mA (Rigaku Corporation, Akishima-shi, Japan). The diffraction angle of patterns was collected from 3.0° to 20.0° with a scanning speed of 3.0°/min.

FTIR spectrum of Na-MT and corresponding organo-MT were collected by using the Nicolet Is 50 fourier infrared spectrometer (Thermo Fisher Scientific, Waltham, MA, USA) with the KBr pellet method (1.00 mg of sample homogenized with 200.00 mg KBr). The wavenumber range was 4000–400 cm$^{-1}$ with resolution of 4.0 cm$^{-1}$.

TG analysis was performed on TG/SDTA851e thermogravimetric/differential thermal synchronization analyzer (Mettler-Toledo, Zurich, Switzerland). The sample was heated from 30.0 °C to 1000.0 °C at a rate of 20.0 °C/min. The flow of high purity nitrogen was 20.0 mL/min.

Contact angle test was performed on DSA 30 optical contact angle analyzer (Kruss, Hamburg, Germany). For this purpose, a pellet was prepared under 5.0–10.0 MPa pressure using about 0.50 g sample.

MD simulation was performed by Materials Studio 7.0 software (Accelrys, San Diego, CA, USA) to further investigate the conformation of alkyl ammonium within organo-MT interlayer. According to the semi-cell structural formula, 4a × 2b × 1c super-cell models were established for simulation study. Our previous study had proved that when the mass of modifier is equivalent to the 1.0 CEC of Na-MT, alkyl ammonium cation can totally enter into the interlayer of Na-MT [29]; therefore, according to the calculation result, we, respectively, added six alkyl ammonium cation into $C_n$-MT1 interlayer and eleven alkyl ammonium cation into $C_n$-MT2 interlayer to ensure total charge balance of the crystal. Then we added different amounts of $H_2O$ to ensure that the $d_{001}$ value of simulation result was consistent with XRD data. The number of alkyl ammonium cation and $H_2O$ added is shown in Table 3.

**Table 3.** The number of alkyl ammonium cation and $H_2O$ added in the interlayer domain of organo-MT.

| Type | $C_{12}$-MT1 | $C_{14}$-MT1 | $C_{16}$-MT1 | $C_{18}$-MT1 |
|---|---|---|---|---|
| $H_2O$ | 110 | 140 | 170 | 219 |
| alkyl ammonium cation | 6 | 6 | 6 | 6 |
| **Type** | **$C_{12}$-MT2** | **$C_{14}$-MT2** | **$C_{16}$-MT2** | **$C_{18}$-MT2** |
| $H_2O$ | 30 | 40 | 50 | 70 |
| alkyl ammonium cation | 11 | 11 | 11 | 11 |

The geometry optimization task was performed to relax the structure of organo-MT model under the following conditions: the organo-MT layers were assumed to be rigid, the cell parameters $a$, $b$, $\alpha$, $\gamma$ remained unchanged, and $c$ and $\beta$ were variable. The algorithm selected was Smart Minimizer, the long-range electrostatic interaction used was Ewald summation method, and the short-range van der Waals used was atom-based summation method. The vacuum cutoff radius was 12.5 Å, the spline width was 1.0 Å, the buffer width was 0.5 Å, and the number of iteration steps was 5000.

The geometry optimization results of organo-MT models are shown in Table 4. After geometry optimization, molecular dynamics simulation was carried out under NPT ensemble and NVT ensemble, respectively. The simulation temperature was 298.0 K, pressure was $1.0 \times 10^{-4}$ GPa, the time step length was 1.0 fs, the numbers of steps were $2 \times 10^6$, and the total simulation time was 2.0 ns. During the simulation, atomic coordinates and lattice parameters were allowed to change freely, and the output data was used for result analysis.

**Table 4.** The geometry optimization results of organo-MT models.

| Type | $C_{12}$-MT1 | $C_{14}$-MT1 | $C_{16}$-MT1 | $C_{18}$-MT1 |
|---|---|---|---|---|
| Volume/Å$^3$ | 7391.34 | 8455.92 | 8784.63 | 13224.80 |
| $c$/Å | 22.6606 | 25.7746 | 29.2132 | 35.5689 |
| $\beta$/° | 97.8919 | 99.8504 | 103.7590 | 101.2330 |
| **Type** | **$C_{12}$-MT2** | **$C_{14}$-MT2** | **$C_{16}$-MT2** | **$C_{18}$-MT2** |
| Volume/Å$^3$ | 7153.53 | 7036.72 | 7849.23 | 8609.17 |
| $c$/Å | 21.3819 | 22.8186 | 24.0655 | 28.6229 |
| $\beta$/° | 98.8988 | 104.471 | 99.0000 | 107.4760 |

Before gel apparent viscosity measurement, we weighed 65.80 g xylene solvent in a 50.0 mL beaker, put the beaker at high speed mixer (GFJ-QA, Jiangyin Shuangye Machinery Co., Ltd., Jiangyin, China), submerged solvent dispersion plate to highly 1/3 volume of the beaker, adjusted the speed of high-speed mixer to 500 r/min, added 6.00 g of organo-MT sample under the condition, stirred for 3 min at 1500 r/min, added 95% ethanol solution of 3.8 mL and stirred for 2 min, and then stirred for 5.0 min at 3000 r/min. Then, we removed the beaker after stopping.

We placed the beaker on the digital viscometer (SNB-2, Shanghai Fangrui Instrument Co., Ltd., Shanghai, China) to test the apparent viscosity of organo-MT gel. Using the No. 3 rotor, we adjusted the viscometer until the groove (immersion mark) on the shaft of the rotor just touches the sample. We slowly moved the beaker on the horizontal surface to ensure that the rotor was located in the center of the beaker. We adjusted the measured speed of the viscometer to 6 r/min and read the number of the display screen after rotating it 10 times.

## 3. Results and Discussion

### 3.1. XRD Analysis

XRD patterns of Ca(Na)-MT and organo-MT prepared by alkyl ammonium with different carbon chain length are shown in Figure 2.

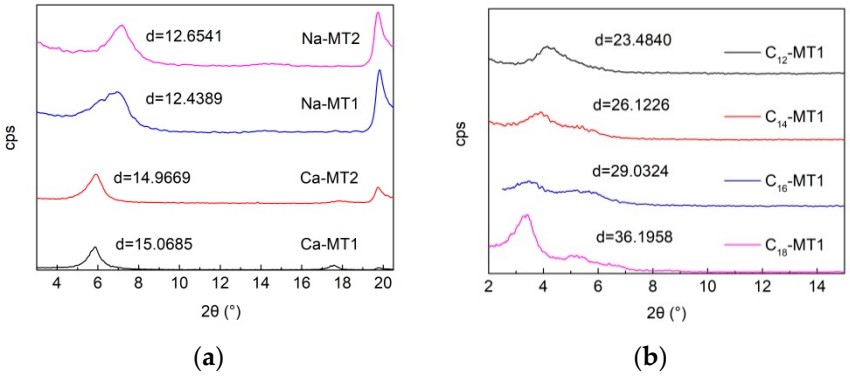

(**a**)　　　　　　　　　　　　(**b**)

**Figure 2.** *Cont.*

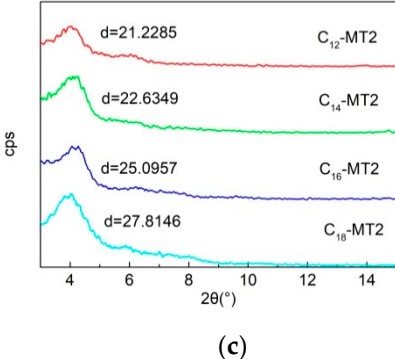

**(c)**

**Figure 2.** XRD patterns of MT, (**a**) Ca(Na)-MT, (**b**) $C_n$-MT1, (**c**) $C_n$-MT2.

Figure 2 shows the XRD patterns of Ca(Na)-MT and organo-MT prepared by alkyl ammonium with different carbon chain length. It can be seen from Figure 2a that the diffraction peaks of (001) of the two kinds of Ca-MT appear at about 5.89°, and the basal spacing $d_{(001)}$ of Ca-MT1 and Ca-MT2 is 15.0685 Å and 14.9669 Å, respectively. After Ca-MT was modified by $Na_2CO_3$, the diffraction peaks of (001) shift towards 7.12°, and the corresponding basal spacing $d_{(001)}$ decreases to 12.4389 Å and 12.6541 Å, respectively. This result proves that Ca-MT have been sodium modified successfully [7].

From Figure 2a,b, we can find that the diffraction peaks of (001) of organo-MT appear at smaller angle than that of Na-MT, which indicates that the interlayer distance of Na-MT is expanded after intercalation of alkyl ammonium [3,30]. Besides, it can be seen that the $d_{(001)}$ value of the two kinds of organo-MT shows stepwise increment with the increase of carbon chain length, for example, the $d_{(001)}$ of $C_{12}$-MT1, $C_{14}$-MT1, $C_{16}$-MT1, and $C_{18}$-MT1 increases from 12.4389 Å to 23.4840 Å, 26.1226 Å, 29.0324 Å, and 36.1958 Å, respectively. The reason is that the size of alkyl ammonium cation increases with the increase of carbon chain length, when Na-MT is modified by alkyl ammonium with long carbon chain, the $d_{(001)}$ of $C_n$-MT1 will evidently increase due to the spatial volume effect. Previous studies also found similar conclusions [11,30,31]. But the $d_{(001)}$ values acquired by us are obviously higher than that obtained by Brito et al. [32], this difference may be mainly caused by different modification method, drying time and layer charge density of Na-MT.

Comparing Figure 2b,c, we can find that the layer charge density of Na-MT can also evidently influence the $d_{(001)}$ values of organo-MT. For the same type of alkyl ammonium, the $d_{(001)}$ value of $C_n$-MT1 is higher than that of $C_n$-MT2. The reason is that the lower the layer charge density of Na-MT, the weaker the electrostatic attraction between $Na^+$ and tetrahedral surfaces, during the process of organic modification, a large amount of alkyl ammonium are absorbed into interlayer of domain, Na-MT1 sheets expand easier along the direction of z-axis due to the spatial volume effect of alkyl ammonium, which results in that the $d_{(001)}$ of $C_n$-MT1 is higher than that of $C_n$-MT2.

*3.2. FTIR Measurement*

FTIR of Na-MT and organo-MT prepared under different conditions is shown in Figure 3.

It can be seen from the Figure 3 that the position of most absorption peaks of organo-MT does not change after the organic modifier is inserted into Na-MT interlayer, which indicates that the intercalation of organic modifier does not destroy the crystal structure of Na-MT. The absorption bands at 3621 $cm^{-1}$ for $C_n$-MT1 and 3619 $cm^{-1}$ for $C_n$-MT2 are assigned to the stretching vibration of $H_2O$ on the surfaces of MT and –OH groups within the octahedral sheets [10]. The broad absorption bands at 3449 $cm^{-1}$ for $C_n$-MT1 and at 3440 $cm^{-1}$ for $C_n$-MT2 are ascribed to the –OH stretching vibration, and the absorption bands at 1638 $cm^{-1}$ for $C_n$-MT1 and at 1639 $cm^{-1}$ for $C_n$-MT2 are corresponding to –OH bending vibration in $H_2O$. The intensity of these characteristic peaks is lower in organo-MT than that in Na-MT, which indicates that the water content of organo-MT interlayer is lower than that in corresponding Na-MT. The reason is that after the intercalation of alkyl ammonium, $Na^+$ are exchanged by hydrophobic alkyl ammonium cation; as a result, the hydrophility of the

prepared organo-MT decreases, which leads to the decrease of the content of $H_2O$ between organo-MT sheets [33]. The absorption bands at 2924 cm$^{-1}$ and 2853 cm$^{-1}$ can be assigned to the $-CH_2$ asymmetric stretching vibration and $-CH_2$ symmetric stretching vibration, respectively [15,20]. The intensity of these characteristic peaks gradually increases from $C_{12}$-MT to $C_{18}$-MT, which also reflects that the hydrophobicity of organo-MT increases with the increase of carbon chain length [30]. The absorption band at about 1478 cm$^{-1}$ is the $-CH_2$ bending vibration. The strong absorption bands near 1033 cm$^{-1}$ for $C_n$-MT1 and 1040 cm$^{-1}$ for $C_n$-MT2 belong to the stretching vibration of Si–O groups of tetrahedral sheets. The change of their shape after intercalation of alkyl ammonium implies that there may be an interaction between Si–O surfaces and alkyl ammonium cation [34]. In addition, the absorption bands at 515 cm$^{-1}$ and 463 cm$^{-1}$ for $C_n$-MT1 and 518 cm$^{-1}$ and 471 cm$^{-1}$ for $C_n$-MT2 are attributed to Mg–O–Si and Mg–O bending vibration, respectively. Apparently, when layer charge density increases, these characteristic bands move towards high frequency region. The reason is that the higher the layer charge density, the more the number of Mg$^{2+}$ in octahedra, and the more obvious the bending vibration of Si–O–Mg and Mg–O in high frequency zone.

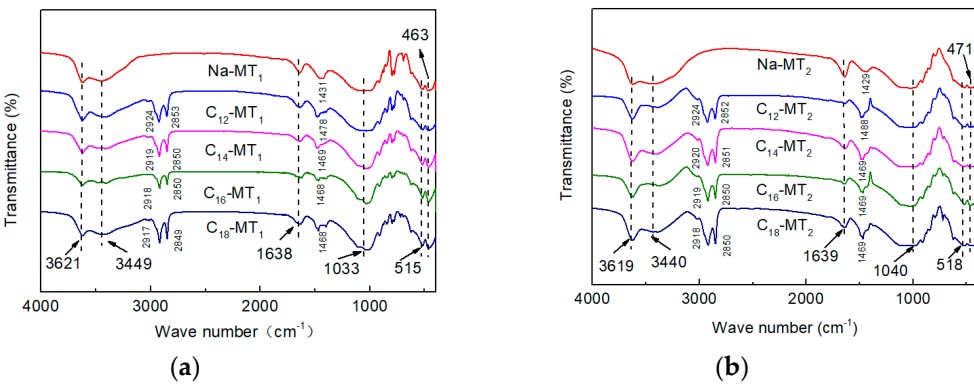

**Figure 3.** FTIR of Na-MT and prepared organo-MT, (**a**) MT1; (**b**) MT2.

### 3.3. TG Experiment

TG curves of Na-MT and corresponding organo-MT prepared under different conditions are shown in Figure 4. The mass loss of organo-MT at different temperature is shown in Table 5.

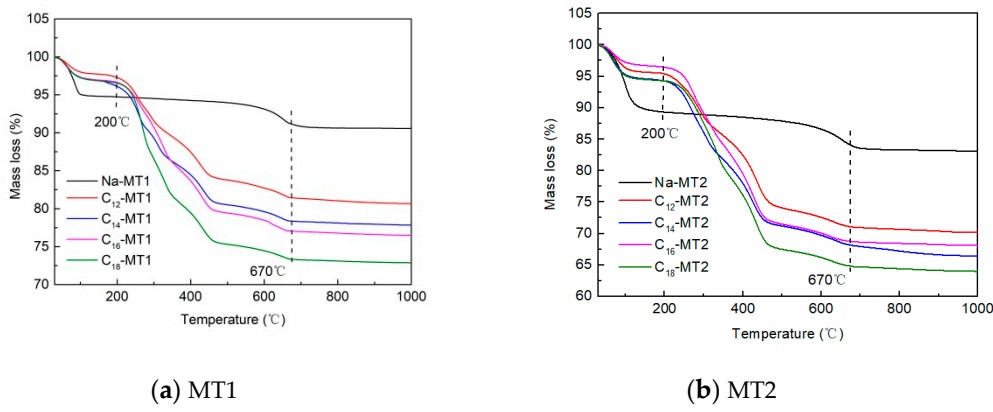

**Figure 4.** Thermogravimetric (TG) curves of Na-MT and prepared organo-MT, (**a**) MT1; (**b**) MT2.

**Table 5.** The mass loss of organo-MT at different temperature range (wt %).

| *T* (°C) | C$_{12}$-MT1 | C$_{14}$-MT1 | C$_{16}$-MT1 | C$_{18}$-MT1 |
|----------|--------------|--------------|--------------|--------------|
| 0–200 | 2.4 | 3.3 | 3.3 | 3.3 |
| 200–670 | 16.0 | 18.3 | 19.5 | 23.4 |

| *T* (°C) | C$_{12}$-MT2 | C$_{14}$-MT2 | C$_{16}$-MT2 | C$_{18}$-MT2 |
|----------|--------------|--------------|--------------|--------------|
| 0–200 | 4.1 | 5.2 | 6.4 | 5.2 |
| 200–670 | 24.8 | 26.5 | 26.6 | 30.0 |

The mass loss of organo-MT below 200 °C is due to that $H_2O$ absorbed in organo-MT pores and layers are released [35]. In this temperature range, the mass loss of organo-MT is lower than that of Na-MT, which further indicates that there are less $H_2O$ within organo-MT interlayer, this phenomenon is consistent with the result of FTIR that the intensity of characteristic peaks at 3449 cm$^{-1}$ for C$_n$-MT1 and at 3440 cm$^{-1}$ for C$_n$-MT2 is low. It can be explained by that the intercalary or adsorbed alkyl ammonium cation reduce the surface energy of Na-MT and change the hydrophilic surfaces to the hydrophobic [6]. The mass loss of organo-MT between 200–670 °C is related to the thermal decomposition of the physically adsorbed alkyl ammonium and intercalated alkyl ammonium cation [16]. In this temperature range, the mass loss between 200 °C and 440 °C is attributed to the oxidation of alkyl ammonium, and the mass loss between 440 °C and 670 °C indicates that the complete oxidation of carbonaceous is formed by previous thermal decomposition [36]. In addition, The mass loss above 670 °C is ascribed to the dehydroxylation of organo-MT [10].

As shown in Figure 5 and Table 5, we can also find that both the carbon chain length of alkyl ammonium and the layer charge density of Na-MT have an evident influence on the mass loss of organo-MT prepared under different conditions, for example, the mass loss of alkyl ammonium in C$_n$-MT1 gradually increases from 16.0% to 23.4%, and the mass loss of alkyl ammonium in C$_n$-MT2 gradually increases from 24.8% to 30.0%. This may be attributed to two factors, on the one hand, SSA of Na-MT2 is higher than that of Na-MT1 (Table 2), which results in that the adsorption capacity of Na-MT2 on alkyl ammonium is higher than that of Na-MT1, on the other hand, layer charge density of Na-MT2 is also higher than that of Na-MT1; thus, more alkyl ammonium are exchanged into Na-MT2 interlayer domain by cation exchange. However, the difference of mass loss between C$_{14}$-MT2 and C$_{16}$-MT2 is not apparent, and the reason may be that Na-MT2 layers cannot be fully dispersed in aqueous solution because of the stronger electrostatic attraction between Na-MT2 sheets, and the pore diameter of Na-MT2 particles is smaller than that of Na-MT1; as a result, the larger alkyl ammonium cation (HTA$^+$) do not fully intercalate into the interlayer of Na-MT2.

### 3.4. Contact Angle Test

In order to further study the influence of layer charge density of Na-MT and carbon chain length of alkyl ammonium on the surface property of organo-MT, we tested the contact angle of different organo-MT, and the results are shown in Figure 5.

From Figure 5, we can see that the contact angle of C$_{12}$-MT1, C$_{14}$-MT1, C$_{16}$-MT1, C$_{18}$-MT1, C$_{12}$-MT2, C$_{14}$-MT2, C$_{16}$-MT2, and C$_{18}$-MT2 is 55.31°, 64.79°, 82.80°, 83.26° 64.81°, 75.11°, 85.84°, and 83.86°, respectively. Apparently, the layer charge density of Na-MT and carbon chain length of alkyl ammonium have significant influence on the surface property of organ-MT, the higher the layer charge density, and the bigger the contact angle of organo-MT. The rule is similar to the analysis results of XRD and TG analysis. This can be explained by that both layer charge density and SSA of Na-MT2 are higher than that of Na-MT1, which results in that the adsorption amount of alkyl ammonium on Na-MT2 surfaces and into Na-MT layers is higher than that of Na-MT1; thus, the hydrophobicity of C$_n$-MT2 is higher than that of C$_n$-MT1. Moreover, either in C$_n$-MT1 or C$_n$-MT2, the contact angle increases with the increase of carbon chain length of alkyl ammonium. The reason is that alkyl ammonium with long carbon chain can cover the organo-MT surfaces better. In addition, it can be

seen from Figure 5 that, when the carbon chain length of alkyl ammonium reaches sixteen (HTAC), the contact angle of $C_{16}$-MT suddenly increases, and it does not change much after that. That is why alkyl ammonium with long carbon chain, such as HTAC and OTAC, are usually selected as organic modifiers in industrial production.

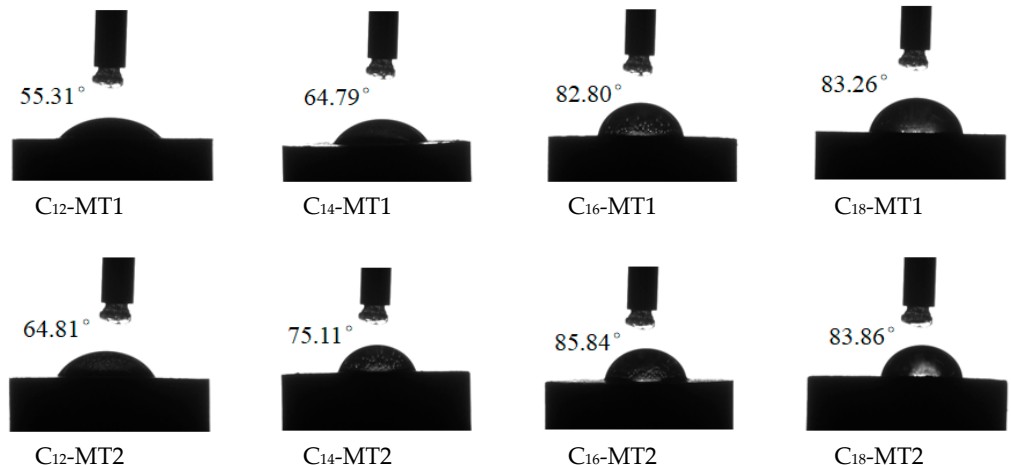

**Figure 5.** Contact angle of organo-MT.

### 3.5. MD Simulation

In order to further investigate the microstructure characteristics of organo-MT interlayer domain, MD simulation method was used to study the z-axis concentration distribution of C and N elements and the arrangement of modifier within organo-MT sheets. The results are shown in Figures 6 and 7.

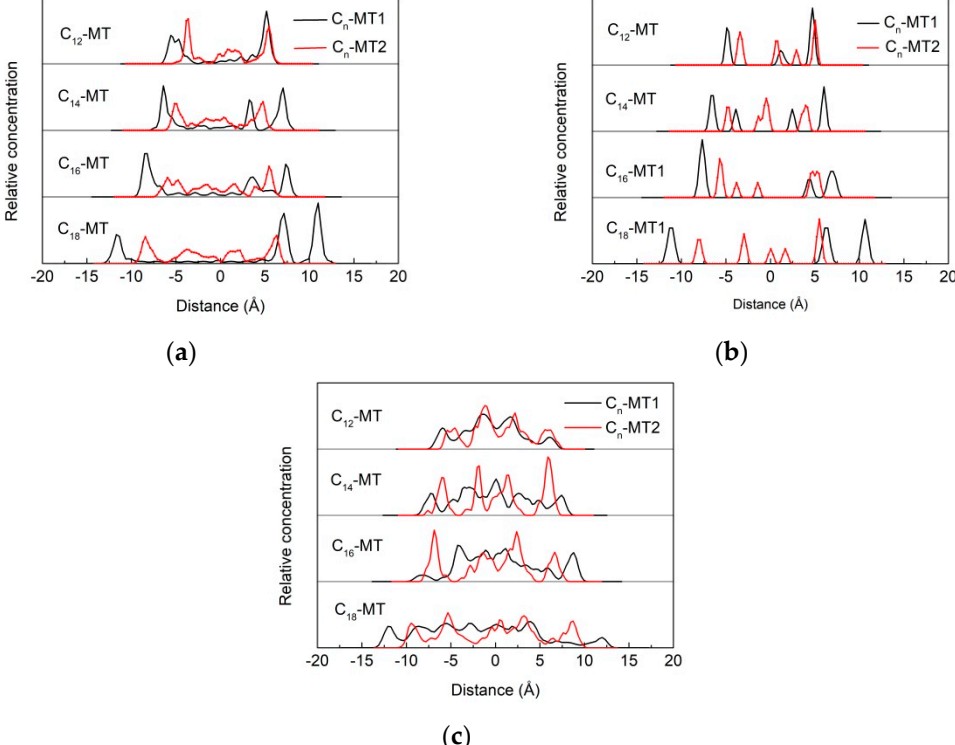

**Figure 6.** The z-axis concentration curves of C, N, and $H_2O$ within organo-MT sheets. (**a**) C, (**b**) N, and (**c**) $H_2O$.

### 3.5.1. Z-Axis Concentration Distribution

It can be seen that the layer charge density of Na-MT and carbon chain length of alkyl ammonium can evidently influence the concentration distribution of C, N, and $H_2O$ within organ-MT interlayer. From Figure 6a, we can find that in $C_n$-MT1, with the increase of carbon chain length, the concentration distribution of C gradually changes from two symmetrical peaks in $C_{12}$-MT1 into three asymmetrical peaks in $C_{18}$-MT1, and the corresponding distribution range increases from −6.5–+6.5 Å to −12.5–+12.5 Å along the direction of z-axis. The reason is that alkyl ammonium with long carbon chain can evidently increase the interlayer distance of Na-MT due to the spatial volume effect (Figure 2), and, because of electrostatic attraction, alkyl ammonium cation gradually moves towards tetrahedral sheets of $C_n$-MT1, which results in that alkyl ammonium is distributed over wider area with the increase of carbon chain length. In $C_n$-MT2, the concentration distribution of C gradually changes from three peaks in $C_{12}$-MT2 into four peaks in $C_{18}$-MT2, and the corresponding distribution zone increases from −4.5–+7.0 Å to −10.0–+7.5 Å.

It can be seen from Figure 6b that the concentration distribution rule of N is similar to that of C, in $C_n$-MT1, the N element exhibits evident double layers distribution in $C_{12}$-MT1, and gradually forms three layers asymmetrical distribution with the increase of carbon chain length, which indicates that some of alkyl ammonium tend to arrange obliquely within $C_n$-MT1 interlayer when the carbon chain length of modifier increases. But when the layer charge density of Na-MT is high, the N element mainly shows three layers distribution in $C_{12}$-MT2 and changes into uneven four layers distribution in $C_{18}$-MT2. The distribution range of N increases from −5.2–+5.2 Å to −12.4–+12.4 Å for $C_n$-MT1, and from −3.8–+5.2 Å to −7.5–+6.2 Å for $C_n$-MT2. Thus, we can conclude that the concentration distribution of C and N in $C_n$-MT2 is closer to tetrahedral surfaces than that in $C_n$-MT1. The reason is that the layer charge density of Na-MT2 is higher, so the polar force between $C_n$-MT2 layers is stronger, which results in that alkyl ammonium cation between $C_n$-MT2 sheets arrange more compact than that in $C_n$-MT1. This simulation results can explain the XRD results well (Figure 2).

In addition, it can be seen from Figure 6c that, in $C_n$-MT1, the concentration distribution of $H_2O$ exhibits four peaks in $C_{12}$-MT1, but their intensity decreases with the increase of carbon chain length, and the distribution range of $H_2O$ increases from −7.5–+7.5 Å to −13.0–+13.0 Å along the direction of z-axis, which indicates that $H_2O$ gradually distribute evenly in interlayer of $C_n$-MT1. When the layer charge density of Na-MT is high, $H_2O$ gradually shows an obvious four layers of distribution as the carbon chain length of modifier increases, and the corresponding distribution range increases from −6.0–+7.5 Å to −10.2–+10.0 Å. The reason is that the concentration distribution of $H_2O$ is influenced by alkyl ammonium and MT sheets, so when the carbon chain length of modifier increases, some of alkyl ammonium tends to move towards tetrahedral surfaces of $C_n$-MT2, which results in that part of $H_2O$ being squeezed and eventually forming an arrangement of four layers.

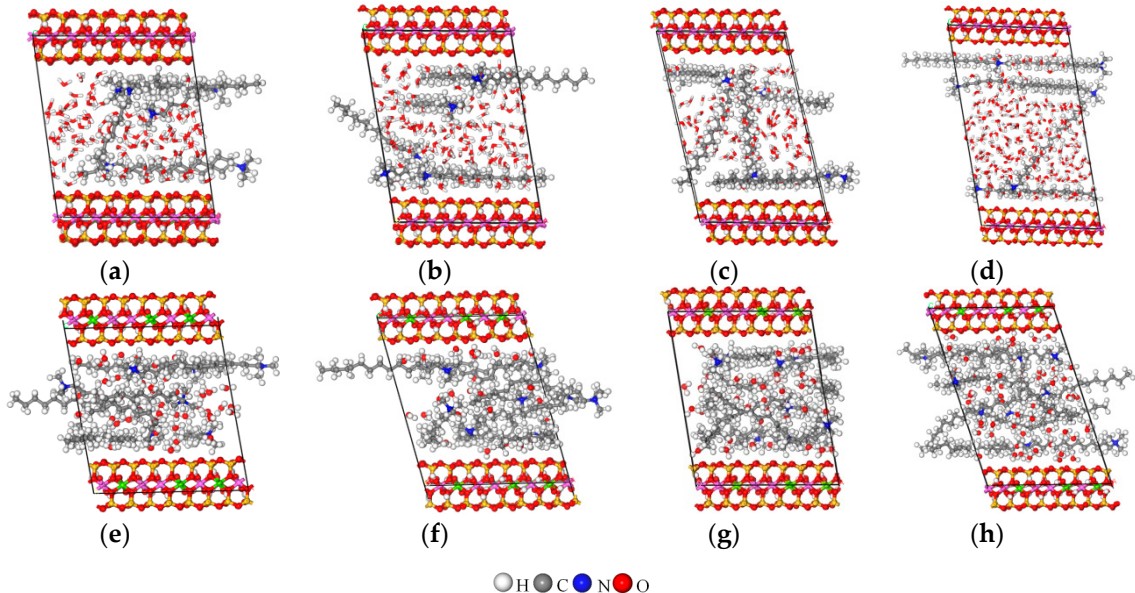

**Figure 7.** The snapshot of alkyl ammonium cation in organo-MT layers. (**a**) C12-MT1; (**b**) C14-MT1; (**c**) $C_{16}$-MT1; (**d**) C18-MT1; (**e**) $C_{12}$-MT2; (**f**) $C_{14}$-MT2; (**g**) $C_{16}$-MT2; (**h**) C18-MT2.

### 3.5.2. Arrangement of Alkyl Ammonium Cation within Organo-MT Interlayer

The snapshots of alkyl ammonium within organo-MT layers are shown in Figure 7, and according the z-axis concentration of C, N, and $H_2O$ within organo-MT interlayer, it can be inferred that, in $C_{12}$-MT1, alkyl ammonium tend to form double layers distribution, and, with the increase of carbon chain length, some of alkyl ammonium are still parallel distributed to $C_n$-MT1 sheets, but others are obliquely distributed within the $C_n$-MT1 sheets, which results in that the concentration of C and N exhibits three asymmetrical peaks (Figure 6). When the layer charge density is high, with the increase of carbon chain length, the distribution of alkyl ammonium gradually changes from three layers into an arrangement of four layers within $C_n$-MT2 sheets.

### 3.6. Gel Apparent Viscosity of Organo-MT

In order to further study the effect of layer charge density of Na-MT and carbon chain length of alkyl ammonium on the application performance of organo-MT, we tested the gel apparent viscosity of different organo-MT in xylene/ethyl alcohol system, and the results are shown in Table 6.

**Table 6.** Gel apparent viscosity of organo-MT (Pa·s).

| Type | $C_{12}$-MT1 | $C_{14}$-MT1 | $C_{16}$-MT1 | $C_{18}$-MT1 |
|---|---|---|---|---|
| Gel apparent viscosity | 1.87 | 2.28 | 3.71 | 13.31 |
| **Type** | **$C_{12}$-MT2** | **$C_{14}$-MT2** | **$C_{16}$-MT2** | **$C_{18}$-MT2** |
| Gel apparent viscosity | 1.20 | 1.92 | 1.60 | 3.49 |

It can be seen that both layer charge density of Na-MT and carbon chain length of alkyl ammonium can evidently influence the gel apparent viscosity of organo-MT. The gel apparent viscosity of $C_n$-MT1 is overall higher than that of $C_n$-MT2. The reason is that, when the layer charge density of Na-MT is low, the electrostatic attraction between Na-MT sheets is weak; in an xylene/ethyl alcohol system, xylene is adsorbed into $C_n$-MT1 interlayer domain due to van der Waals interaction force, and, because of the spatial volume effect, large amounts of $C_n$-MT1 sheets are gradually separated, and, as is known to us,

the end face of Na-MT sheets are positively charged, and the interlayer planes are negatively charged, so the stripped $C_n$-MT1 sheets can form house-of-cards structure easily, which leads to the result that the gel apparent viscosity of $C_n$-MT1 is higher. But, in $C_n$-MT2, although large amounts of alkyl ammonium and xylene are adsorbed into the interlayer domain, it is difficult to strip $C_n$-MT2 sheets and form house-of-cards due to its strong electrostatic attraction, which results in the gel apparent viscosity of $C_n$-MT2 being lower than that of $C_n$-MT1.

In addition, from Table 6, we can also find that the gel apparent viscosity of organo-MT in xylene/ethyl alcohol system increases with the increase of carbon chain length. This phenomenon is more obvious when the layer charge density is low ($C_n$-MT1) and is also consistent with the contact angle test result. The reason is that when the alkyl ammonium with long carbon chain enters into Na-MT1 interlayer by cation exchange, and, because of the spatial volume effect, $C_n$-MT1 sheets gradually expand along the direction of z-axis (this phenomenon can also be seen in Figures 2 and 7), and the electrostatic attraction between $C_n$-MT1 layers decreases as the distance between the layers increases; therefore, in the xylene/ethyl alcohol system, when large amounts of xylene are adsorbed into $C_n$-MT1 interlayer, $C_n$-MT1 sheets are more easily separated and form a house-of-cards. As a result, the gel apparent viscosity of $C_n$-MT1 prepared by alkyl ammonium with long carbon chain is higher.

## 4. Conclusions

In the work, the effect of layer charge density of Na-MT and carbon chain length of alkyl ammonium on the structure and gel property of organo-MT was systematically studied by using many characterization techniques. The result shows that the interlayer distance of organo-MT increases with the increase of carbon chain length, and when the layer charge density is low, this phenomenon is more obvious. The characteristic absorption bands at 2924 cm$^{-1}$ and 2853 cm$^{-1}$ appear in FTIR decrease with the increase of carbon chain length, and, when the layer charge density increases, the characteristic peaks at 515 cm$^{-1}$ and 463 cm$^{-1}$ shift towards high frequency region. The mass loss of organo-MT with high layer charge density is larger than that of organo-MT with low layer charge density, and, when carbon chain length of modifier increases, the mass loss of organo-MT also increases. In addition, the contact angle of $C_n$-MT2 is higher than that of $C_n$-MT1 when the carbon chain length of modifier is the same, and alkyl ammonium with long carbon chain can significantly improve the hydrophobicity of Na-MT. MD simulation results show that the concentration distribution of C and N in $C_n$-MT2 is closer to tetrahedral surfaces than that in $C_n$-MT1. In $C_n$-MT1, alkyl ammonium cation forms obvious double layers when the carbon chain length of modifier is short, and, with the increase of carbon chain length, some of alkyl ammonium tend to be tilt with $C_n$-MT1 surfaces. In $C_n$-MT2, the arrangement of alkyl ammonium gradually changes from three layers into four layers with the increase of carbon chain length. The gel apparent viscosity of organo-MT in xylene/ethanol system increases with the increase of the layer charge density of Na-MT or the carbon chain length of alkyl ammonium.

**Author Contributions:** Conceptualization, J.Q.; methodology, D.L.; software, D.L. and Y.W. (Yueting Wang); validation, S.J., G.C., G.W. and X.L. (Xianjun Lyu); formal analysis, G.L., Y.W. (Yaqi Wang) and W.W.; investigation, D.L., Y.W. (Yueting Wang) and P.W.; Resources, X.L. (Xianjun Lyu); data curation, D.L., S.J. and X.L. (Xiaodong Liu); Writing—original draft preparation, D.L.; Writing—review & editing, J.Q., Y.W. (Yueting Wang) and G.C; visualization, J.Q.; supervision, G.W.; project administration, X.L. (Xianjun Lyu); funding acquisition, J.Q. and G.W. All authors have read and agreed to the published version of the manuscript.

**Funding:** This work was supported by the National Nature Science Foundation of China—"Design of structure and gelling performance of Montmorillonite/Alkylammonium based on the adsorption properties of Alkylammonium on Montmorillonite (No: 51774200)" and "Study on the immobilization effect and mechanism of heavy metal in sulfide-mine tailings by chelating agent modified bentonite (No: 51764003)".

**Conflicts of Interest:** The authors declare no conflict of interest.

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
