# Peer review of "Comprehensive Characterization of the Structure and Gel Property of Organo-Montmorillonite: Effect of Layer Charge Density of Montmorillonite and Carbon Chain Length of Alkyl Ammonium"

_minerals, doi:10.3390/min10040378_

Round 1

Reviewer 1 Report

Good paper, well writen. I would suggest to change  cetyl trimethyl ammonium chloride (CTAC) to hexadecyl trimethyl ammonium chloride (HTAC). Please pay more attention to proper writing in References, for example number 31:

https://www.sciencedirect.com/science/article/pii/S0021979716303678?via%3Dihub

Author Response

Dear reviewer,

Thank you very much for the critical feedback and the valuable comments.

We have studied your comments carefully and have made some revisions which are marked in yellow in the paper. We have tried our best to revise our manuscript according to the comments.

Please refer to the attachment for details.

Reviewer 2 Report

Jun Qiu et al. have studied the effect of layer charge density on the structure of alkyl ammonium molecules with different carbon chain length. Overall, this is an interesting story and worth publishing. The authors refer to many different experimental and theoretical technologies to underline their results. However, some things could be enhanced: (1) Starting in the abstract, the authors mix adsorption and absorption - which are completely different things. This word-mixing needs to be corrected over the complete manuscript. (2) Since the authors here start with material from two different regions, how can they exclude that the seen effects are a results of discrepancies in the structure of the particles? (3) Some of the data, like the IR, should be better compared to existing models in literature. There are models, making it possible to calculate the exakt angle of the carbon chain standing to the clay surface. Can this be applied here? (Langmuir 2010, 26, 1, 156-164)

Author Response

Dear reviewer,

Thank you very much for the critical feedback. We felt lucky that our manuscript was sent to you.

We have studied your comments carefully and have made some revisions which are marked in green in the paper.

please refer to the attachment for details.

Reviewer 3 Report

The manuscript presented by Jun Qiu et al, which is discussing characterization of alkyl-ammonium intercalates in montmorillonites, is clearly of interest for the mineralogical community. Although alkyl-ammonium intercalates in montmorillonites has been studied for decades, this study provides interesting multi-method analysis for montmorillonites with different layer charges. In wide range of publications only one type of smectite is studied, while in this case this is a comparative study, which is worth emphasizing.

Overall, I feel that the readers of the Minerals would find the observations noted in the article of interest, and I would recommend publishing the article following minor revision.

1) There is disagreement between experimental and MD results. Based on IR and TG analyses it is clear that amount of water is much smaller than that for pure Na-MT. Based on Figure 4 it is 20-50% of that for Na-MT. Other MD simulations shows that in the case of 1 water layer (12.5A) it is about 5 H2O molecules per unit cell. Therefore, in your intercalates you should assume ca. 1-3 H2O molecules per unit cell (8-24 H2O per 8 unit cells of your system). While you assume even 219 H2O molecules, which corresponds to 5 water layers. It is clearly not structure that corresponds to your experimental results. I would suggest keeping the results that you already shown but adding simulations with correct number of water molecules but higher number of alkylammonium chlorides (you need to include chlorine in your simulations).

2) The simulation cell is too small, as single alkylammonium molecule can interact with itself. I suggest considering at least 6a x 3b x 1c unit cells. This should however not affect the received results significantly.

3) Figure 2 b: there is significant difference in received d001, which is not visible in diffractograms: peak positions are practically identical, while 23A should be at about 4deg 2theta, 29A at 3deg 2theta, and 36A at 2.5deg 2theta. Maybe you do not have such significant differences in d001 and therefore you can assume much smaller number of water molecules in MD. Please check it.

4) Paragraph 3.5. should be reworked after addressing points 1 and/or 3, as there can be different basal spacing or amount of water for MT1.

Smaller points:

- l. 19 – MT abbreviation is not explained.

- please note if you measured humidity in the laboratories as these can affect the received XRD, IR and TG results.

- l. 189 and following – precision of d001 measurements is much smaller: around two significant digits maximally.

- l. 218-219: bands at 3621 and 3619 corresponds both to OH groups and water molecules on the surface of smectite.

- l. 234 – another explanation can be: lack of interactions with Na+ ions, not necessarily interaction between Si-O surfaces and alkyl ammonium cation.

- l. 238: in octahedra (plural).

- Figure 4 – what is source of mass decrease at T>660C for C14-MT?

- l. 254 – 255 – what is difference between ranges 200-440 and 440-670 oC – this is unclear. Please be more specific.

- Figure 6 – please add also distribution of water molecules.

Author Response

Dear reviewer,

Thank you very much for the critical feedback. We feel lucky that our manuscript is sent to you.

We have studied your comments carefully and have made some revisions which are marked in red in the paper.

please refer to the attachment for details.
